# The Impact of Liquids and Saturated Salt Solutions on Polymer-Coated Fiber Optic Sensors for Distributed Strain and Temperature Measurement

**DOI:** 10.3390/s24144659

**Published:** 2024-07-18

**Authors:** Martin Weisbrich, Dennis Messerer, Frank Holzer, Ulf Trommler, Ulf Roland, Klaus Holschemacher

**Affiliations:** 1Structural Concrete Institute, Leipzig University of Applied Sciences (HTWK Leipzig), 04275 Leipzig, Germany; dennis.messerer@htwk-leipzig.de (D.M.); klaus.holschemacher@htwk-leipzig.de (K.H.); 2Department of Technical Biogeochemistry, Helmholtz Centre for Environmental Research (UFZ), 04318 Leipzig, Germany

**Keywords:** DFOS, distributed fiber optic sensing, fiber optic sensor, fiber coating, strain measurement, temperature measurement, SHM (Structural Health Monitoring)

## Abstract

The application of distributed fiber optic strain and temperature measurement can be utilized to address a multitude of measurement tasks across a diverse range of fields, particularly in the context of structural health monitoring in the domains of building construction, civil engineering, and special foundation engineering. However, a comprehensive understanding of the influences on the measurement method and the sensors is essential to prevent misinterpretations or measurement deviations. In this context, this study investigated the effects of moisture exposure, including various salt solutions and a high pH value, on a distributed strain measurement using Rayleigh backscattering. Three fiber optic sensors with different coating materials and one uncoated fiber were exposed to five different solutions for 24 h. The study revealed significant discrepancies (∼38%) in deformation between the three coating types depending on the surrounding solution. Furthermore, in contrast to the prevailing literature, which predominantly describes swelling effects, a negative deformation (∼−47 με) was observed in a magnesium chloride solution. The findings of this study indicate that corresponding effects can impact the precision of measurement, potentially leading to misinterpretations. Conversely, these effects could be used to conduct large-scale monitoring of chemical components using distributed fiber optic sensing.

## 1. Introduction

Distributed fiber optic strain and temperature measurement is a promising method for monitoring and detecting potential damage in a variety of applications, particularly in civil engineering [1,2,3,4]. Brillouin and Rayleigh backscattering methods permit precise and continuous measurement of strain and temperature along optical fibers applied to the surface or inside structures to be monitored [5,6,7]. This technology offers significant advantages over conventional sensors, including high spatial resolution and long-range measurements [8,9]. In addition, fiber optic sensors are immune to electromagnetic radiation and their long-term stability is of particular interest for structural health monitoring (SHM) in civil engineering [10]. The specific characteristics of both methods have been described in numerous publications, including [7].

One of the advantages of distributed fiber optic sensing (DFOS) is the ability to measure deformations and temperatures within the building material matrix. The small size of the sensors allows their integration into almost any matrix without affecting its structural behavior. This enables direct assessment of the mechanical and thermal response of materials, thereby opening up a wide range of applications. Previous studies demonstrated the efficacy of DFOS in measuring deformation and temperature in concrete and mortar [7,11,12,13]. Moreover, as shown in [14,15,16,17], crack detection and crack width determination are also possible. Furthermore, DFOS has also been used to measure the drying shrinkage of cement matrices, yielding results with a low degree of discrepancy relative to the reference measurement [11].

For the purpose of utilising DFOS in the context of measurement tasks, such as SHM, a comprehensive understanding of the sensor behaviour is of significance. Fiber optic sensors are composed of a glass fiber encased in a polymer coating. In the context of construction applications, sensors are typically required to be more robust in order to prevent damage and fiber breakage. Consequently, they are frequently encased with supplementary coatings, jackets, or other protective layers. A number of publications already discussed the structure of optical fibers and the influence of coating layers on the strain behavior [9,11,18,19,20,21]. In [19] it was shown that in addition to the thickness and stiffness of the coating material, the embedding length within the material matrix also affects the strain measurement. Moreover, several studies have identified an other phenomenon influencing both strain and temperature measurements through Rayleigh or Brillouin backscattering: the swelling behavior of coating materials following exposure to humidity [22,23,24,25,26].

In particular, acrylate polymers tend to swell when exposed to moisture [27]. It is also known that pH, temperature, and salt content can influence the swelling effect [28]. In [29,30], this effect was investigated with ethanol, isopropyl alcohol, and acetone in order to be able to measure chemical components in the environment with DFOS. The test results demonstrate that as the concentration of ethanol, isopropyl alcohol, or acetone increases, the swelling effect also increases, which in turn can have a direct effect on the strain signal.

The solvents ethanol, isopropyl alcohol, and acetone play a minor role in the localization of chemical compounds in structural health monitoring. For metrology, however, it is critical to note that the measured strain or temperature can be falsified by the presence of moisture, which can lead to misinterpretation or inaccuracy. This would necessitate subsequent calibration and adjustment. Moreover, the impact of other substances on the swelling effect is not yet fully understood.

In this context, the impact of pertinent substances within concrete structures has been expanded as part of this study. As evidenced in [28], in addition to temperature and pH, the salt content can exert a significant influence on the swelling effect. Consequently, distinct salt solutions, deionized water, and a solution extracted from a cement were used in the present study to quantify the strain with DFOS based on Rayleigh-Backscatter OFDR [31]. In addition to different coatings, an uncoated fiber was used as a reference. Moreover, it is important to consider the role of preconditioning in determining potential effects. Finally, with regard to SHM, it is particularly important to determine whether these effects have a hysterese. In particular, during the observation period, the deformation effect detected on the acrylate fiber exhibited a hysteresis phenomenon, which is defined as a phenomenon whereby the response of a system to a given stimulus varies with the previous history of the system.

The results of these investigations could not only lead to a significant increase in measurement reliability, but also provide details for new guidelines for the use of fiber-optic strain and temperature measurement systems in moisture-exposed environments. In particular, when monitoring unprotected concrete infrastructure, moisture exposure could falsify strain and temperature measurements and potentially lead to misinterpretation. On the other hand, spatially resolved moisture detection would be possible for integrated fiber optic sensors if the coatings were adapted accordingly.

## 2. Experimental Program

### 2.1. Experimental Design

In order to investigate the influence of different liquids on the swelling and shrinkage of the coating material and a possible resulting deformation signal, a fiber optic sensor was placed in a glass basin containing the observed solution (Figure 1). A new, untested fiber was utilized for each liquid. The sensors were installed without tension or any expansion restrictions and measured using Rayleigh backscattering with a spatial resolution of 2.6 mm. Strain values were recorded at 1 Hz for 5 s every 10 min immediately after immersion. To fix the fiber, two 24 G needles were glued to an acrylic glass plate at a distance of 10 cm. The plate covers the entire glass basin to prevent excessive evaporation of the liquids. Prior to immersion, the distance between the fiber sensor at the inflection point and the bottom of the basin was set to 5 mm in an empty basin using calipers. Following immersion, a check was conducted to ensure that the fiber did not touch the bottom of the basin, thus potentially influencing the measurement result. The distance between the acrylic glass plate and the liquid surface was maintained consistently throughout all tests. The fiber sensor was immersed in the liquid over a length of 10 cm. New fiber sensors were used for each solution to avoid the effects of potential hysteresis effects.

The tests were conducted in near-identical environmental conditions of 20 °C and 65% RH. To ensure precise monitoring and control of temperature and humidity, a variety of sensors were employed to record ambient conditions [32,33,34]. The primary objective was to minimize fluctuations in ambient conditions to the greatest extent possible, considering that strain measurement is influenced by temperature change (approx. 10 με/K for the observed sensors). Additionally, the results of this study indicate that the strain change due to moisture-induced swelling or shrinkage of the coating may be contingent on preconditioning. For all tests, the mean standard deviation of the temperature and relative humidity was 0.05 K, and 1.5% RH. The chronological sequence of the tests is illustrated in Figure 2.

### 2.2. Material

Five distinct test liquids were used to assess the impact on the coating material (see Table 1). In addition to deionized water as a reference, a filtered cement solution (667 g/L) was subjected to testing. The filtration process was conducted using folded filter papers [35], resulting in a pH value of 12.6 for the cement solution. Furthermore, three different saturated salt solutions were evaluated. These included sodium chloride, sodium nitrite, and magnesium chloride. The mixing ratios for the saturated solution can be found in Table 1. Throughout the experiment, care was taken to ensure that the liquids, especially the magnesium chloride solution, were always supersaturated to ensure repeatability.

Four different fiber types were used as sensors. For strain measurement applications, the coating material and coating thickness are of great importance, as they directly influence the strain transfer. The fiber coated with ORMOCER^®^ (organically modified ceramic, [36]) was selected, as previous studies have demonstrated the excellent suitability of the coating material for strain measurement [7,9,11,12,13]. The acrylate- and polyimide-coated fibers were selected as exemplars for standard and high-temperature applications, respectively. For comparative purposes, an uncoated SMF-28e+ standard fiber was also investigated. The strain coefficient was determined during tensile tests on all fiber types in previous investigations. Table 2 lists the main characteristics of the sensor used.

## 3. Results

Following the completion of the test, the measured strain values were processed accordingly. In addition to an algorithm for detecting and replacing potential measurement deviations [9,11], the values within the evaluation length (same for all fibers) were selectedand used for further processing. Due to the potential for edge effects, the evaluation length is 50% shorter than the immersion length (Figure 1). Following the viewing and visualization of the data, the values within the evaluation length were averaged in the fiber direction.

Figure 3 presents the results of the strain measurement of the ORMOCER^®^-coated fiber (OCF) in all five observed liquids. Within the cement solution, deionized water, and sodium chloride, positive deformations (swelling) of approximately 67 με, 64 με and 18 με, respectively, were observed. In the case of sodium nitrite, no change in deformation occurred. Immersion in the supersaturated magnesium chloride solution resulted in a negative deformation (shrinkage) of approximately −44 με. After four hours, no further change occurred, and the strain remained constant for the remainder of the observation period. Upon removal of the sensors from the liquids, the strain value returned to the initial state measured after preconditioning.

The results of the polyimide-coated fiber (PCF) are presented in Figure 4. The observed behavior is comparable to that of the OCF, but with significantly lower strains, approximately 35% (Table 3). Even the minor discrepancies between DIW and CEM utilizing OCF (3 με) could be discerned between these two solutions through the use of PCF (2 με). A notable difference between the OCF and PCF can be observed within the magnesium chloride solution: a faster increase in shrinkage. The constant curve is reached after less than 60 min. Additionally, the non-measurable change in the strain value within the sodium nitrite solution is identical to that observed in the OCF. The PCF also demonstrates an immediate decrease in deformation following the extraction of all sensors from the observed liquids.

The results of the acrylate-coated fiber (ACF) show a significantly different strain behavior compared to the fibers with ORMOCER^®^ and polyimide coating (Figure 5, Table 3). While NaCl, DIW, and CEM also result in a positive strain value, permanent negative strains can be observed after extraction. Furthermore, these three sample liquids exhibit a continuous decrease after a strain peak. In the case of DIW and CEM, a decrease in negative strains can be observed. As with the PCF, the initial maximum strain is reached earlier than in the other liquids. After 24 h of immersion, the negative strains in MgCl_2_ show a slight decrease. A complete decrease in strain can be measured after removing the fiber sensor from the saturated magnesium chloride solution.

Figure 6 illustrates the outcomes of all liquids evaluated with the uncoated fiber (UCF). There is minimal variation in strain, both after immersion and after removal of the measuring fiber from the liquids (Table 3). The slight fluctuations in strain are likely attributable to temperature and humidity control and within the range of measurement noise.

Figure 7 and Figure 8 show the comparison of all four fiber measurements within the liquids with the highest amplitude (DIW: positive deformation, MgCl_2_: negative deformation). The three coated fiber types exhibit nearly identical increases and decreases in strain immediately after immersion and after removal. Furthermore, Figure 7 illustrates the decline in strain described above for the ACF in comparison to the other two coated fibers, both after immersion and after removal. The UCF exhibits a slight discrepancy of <5 με in comparison to the zero measurement.

In the MgCl_2_-solution, all three coating materials exhibit negative strains (Figure 8). Both the tests with the OCF and PCF demonstrate consistent curves after approximately 3 h. With the PCF, the area of maximum strain is reached earlier than in the test with the OCF. The maximum range also appears to be reached earlier with the ACF after approximately 60 min, which is confirmed by the tests in the other liquids (Figure 7). After extraction, the strains return to the initial state for all coating materials. A slight delay can be observed with the ACF in comparison to the OCF and PCF.

## 4. Discussion

In conclusion, the measurements of the various coatings exhibit a consistent pattern. The difference between the OCF and the PCF is between 33% to 38%, with the exception of the solution NaNO_2_. The maximum amplitude values of all three coating materials are similarly comparable (Table 2), which reinforces the overall reproducibility. However, certain measurement fluctuations and measurement noise of different origins can also result in a certain degree of measurement deviation in this regard. In particular, the measurement noise in the present study is complex. As previously mentioned in the article, the regulation of temperature and humidity within the climate chamber also contributes to the noise level. Despite the low standard deviation, fluctuations and offsets cannot be completely avoided (e.g., NaNO_2_, Figure 6. Moreover, other factors may contribute to these variations. For instance, the time of taring may have occurred before immersion in a control window. In addition, the following essential sources of measurement noise are present with regard to the fiber optic measurement method: the method itself (light source, detector, backscatter spectrum, spatial resolution, measurement rate), the splice quality (fiber connector, termination), the termination quality, the fiber quality (variations in refractive index and attenuation due to the manufacturing process, Table 2), and the end face quality (fiber connector, bulkhead). Furthermore, if the fiber is immersed in the liquid, there are also potential imperfections that cannot be excluded, such as saturation, temperature stability/field, and evaporation. Additionally, it is unclear whether the deformation effect caused by exposure to moisture can result in local differences in the coating and whether these are completely reversible after removal from the liquid. In the case of OCF, despite the utmost care, there is a possibility that coating residues may remain on the sensor during decoating.

The results presented in Section 3 demonstrate that the deformations of the PCF and OCF observed after immersion are almost completely reversible. For the ACF within the solutions DIW, CEM, and NaCl, a hysteresis is discernible, which is not fully reversible, at least within the measured time periods of 24 h and 48 h (Figure 5). It remains to be determined whether this reversibility is still present after several immersion-extraction cycles, or, in the case of ACF, whether the hysteresis is further increased.

Furthermore, the test results and the deviation from the measured values in DIW, as ascertained by [29,30], indicate that preconditioning may exert a significant influence on both the amplitude and the type of deformation. However, the potential for preconditioning with higher moisture contents or temperatures to result in positive deformation in the case of magnesium chloride, for instance, remains uncertain. It is also probable that preconditioning exerts a non-linear influence on the process and, as a consequence, on strain and temperature measurements. In general, further investigations by [37,38] demonstrate that thermodynamic changes influence the swelling effect.

In [28], it is claimed that the pH value can influence the swelling of polymers within solutions. However, this could not be confirmed based on the results. A slight deviation in the measured strains between CEM (pH 12.6) and DIW of 2 με to 3 με was observed, but this could also be attributed to general measurement fluctuations.

For all liquids, the OCF showed a significant time delay after immersion to reach maximum deformation compared to PCF and ACF. The incorporation of the liquid into the polymer structure appears to take longer. After removal, however, a similar evolution was observed as for PCF.

Although there is variation in the amount of deformation in different solutions (as shown in Figure 3, Figure 4 and Figure 5), there is a general pattern in the order of strain deviation in response to different liquids when the fibers are similarly preconditioned in air. The highest positive strains were observed in DIW and CEM, followed by the NaCl and NaNO_2_ solutions. In contrast, the MgCl_2_ solution exhibited negative strains for all coating materials. This dependency can also be interpreted as an increasing influence of the salt solutions relative to the reference condition of placement in distilled water (DIW). This remarkable finding suggests that the measured strain variations are governed by more fundamental processes that are independent of the type of coating observed. It is hypothesized that this strain effect is related to the affinity of the materials for the salt solution and the resulting enhancement of water in the coating layer.

The order of the strain deviation (DIW > NaCl > NaNO_2_ > MgCl_2_) correlates with the relative humidity determined over the saturated electrolyte solutions in the equilibrium state. The relative humidities are 75%, 66% and 33% for NaCl, NaNO_2_ and MgCl_2_, respectively [39,40,41,42]. For CEM, no corresponding value for relative humidity could be found in the literature. Therefore, it can be postulated that the phase transition of water molecules from the bulk liquid phase into the polymer coatings is controlled by analogous principles as for the transition of water from the liquid to the gaseous phase. Consequently, the activity of water molecules is reduced by the addition of ions, which can be described by Raoult’s law [43,44]. In the case of MgCl_2_, the direction of the phase transition is reversed, resulting in the transport of water molecules absorbed during preconditioning from the polymer to the electrolyte.

When polar polymers, especially polyacrylates, are exposed to water, hydrogels can form. These are also known as superabsorbent polymers. They are used in a variety of applications, including the self-sealing of pipes and the production of water-absorbing fillers [45,46,47]. According to the literature, the degree of swelling depends on two factors: the degree of cross-linking within the polymer and the salt content of the absorbent [46]. The degree of swelling is inversely proportional to the concentration of cations shielding the stationary anions in the polar polymer [48]. This results in a lower strain for electrolyte solutions compared to DIW.

Furthermore, the Donnan theory [47] states that the higher the charge density of the cations in the electrolyte solution, the lower the swelling pressure in the hydrogel. As an illustrative example, a comparison can be made between saturated MgCl_2_ and NaCl solutions. In these solutions, the charge density of Mg^2+^ is significantly higher than that of Na^+^, resulting in a significantly lower swelling pressure. The difference in the degree of strain deviation observed in the ACF for the two sodium salt solutions can be attributed to their disparate saturation concentrations (5.5 mol/L for NaCl and 11.9 mol/L for NaNO_2_). According to Donnan’s theory, the swelling pressure is increased in the NaCl solution due to its lower charge density.

## 5. Conclusions

The results of the conducted research indicate a significant need for further investigation into the effects of environmental conditions on the coating and jacket during fiber optic temperature and strain measurements using Brillouin or Rayleigh backscattering. For all measurement tasks involving changes in humidity, the findings described above should be taken into account for the evaluation of measured deformations and temperatures. Without this consideration, there is a risk of misinterpretation, particularly in the SHM of critical infrastructure. The authors suggest installing a strain decoupled fiber for moisture or chemical compensation in addition to a fiber for strain measurement and temperature compensation. Furthermore, a fiber sensor with a metallic coating could also be used for strain measurement to avoid these influences.

However, the results also indicate that environmental monitoring and moisture measurement may potentially be possible with the use of appropriate coating materials. In [29,30], the concept of detecting acetone, ethanol, or isopropanol has already been proposed. In order to accurately measure moisture or detect chemical components in liquids, the coating materials may need to be tested and adjusted for their response to the particular liquid. Moreover, mechanical and thermal decoupling is essential to prevent cross-effects from strain and temperature. In this regard, DFOS has significant potential for detecting chemical components over large areas with high precision.

In any case, the test results from this study demonstrate a need for further research. Potential effects of cyclic moisture exposure on the fibers have already been discussed. Additionally, there is a need to investigate how the process proceeds at lower salt concentrations and to determine the salt concentration at which the influence becomes negligible. For SHM, investigations on the effects on robust fiber optic cable sensors or other coating and jacket materials are of great importance. Furthermore, the utilisation of repeated measurements may facilitate the attainment of statistical certainty. The adaptation of preconditioning could also provide further insights into the effects mentioned. In this respect, thermodynamic changes, as described in [37,38], could have a significant influence on the strain and temperature measurement. Finally, the results indicated that pH had a variable effect on the outcome and further research is needed in this area.

The authors summarize the following conclusions:The ambient humidity and the concentration of various salts exert a significant influence on the swelling and shrinkage of the polymer coating material of optical fibers.This results in a notable impact on the outcomes of a fiber optic strain or temperature measurement, which is conducted through the use of Brillouin or Rayleigh backscattering.In the case of the ORMOCER^®^- and polyimide-coated fibers, these effects were largely reversible. In the case of the acrylate-coated fiber, hysteresis was observed in deionized water, cement solution, and NaCl solution.The potential for hysteresis during long-term or repeated exposure remains uncertain with regard to the ORMOCER^®^- and polyimide-coated fiber.The preconditioning (humidity and temperature) appears to significantly influence the deformation effect of fiber coatings.Further research is required to investigate the effects of preconditioning, salt concentrations, and climatic conditions, as well as the variation of coating materials.

## Figures and Tables

**Figure 1 sensors-24-04659-f001:**
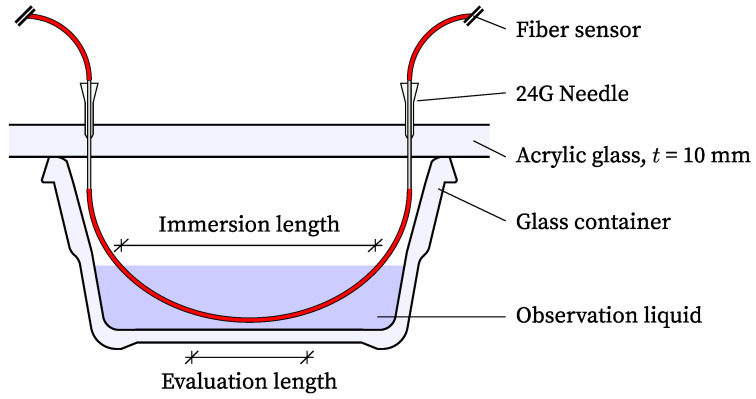
Schematic test setup.

**Figure 2 sensors-24-04659-f002:**
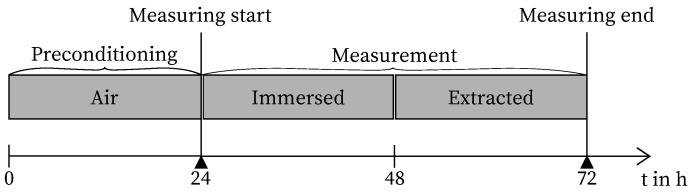
Experimental procedure.

**Figure 3 sensors-24-04659-f003:**
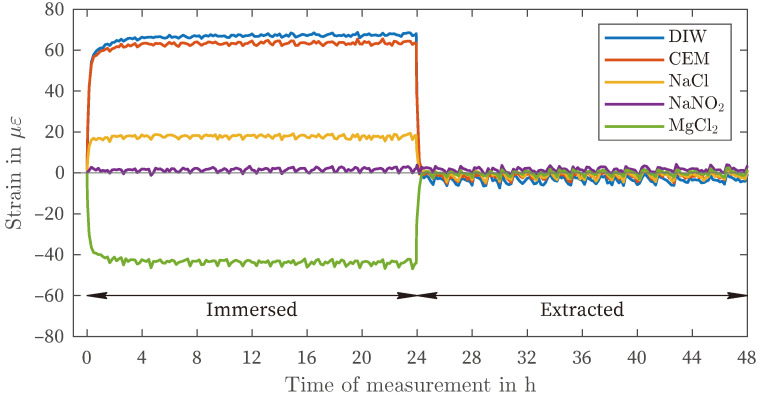
Fluid exposure, ORMOCER^®^-coated fiber.

**Figure 4 sensors-24-04659-f004:**
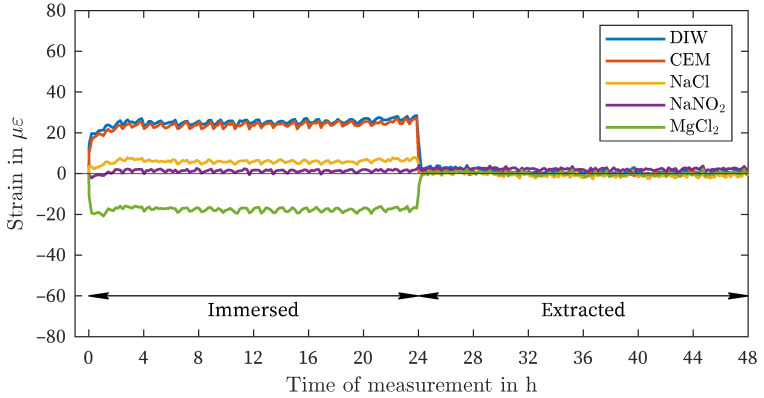
Fluid exposure, Polyimid-coated fiber.

**Figure 5 sensors-24-04659-f005:**
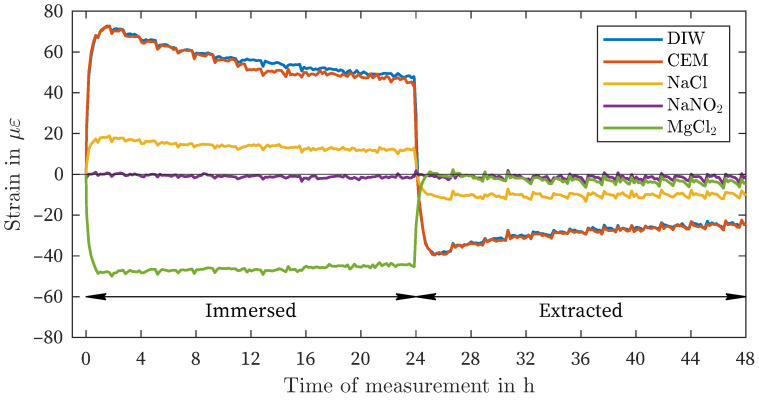
Fluid exposure, Acrylate-coated fiber.

**Figure 6 sensors-24-04659-f006:**
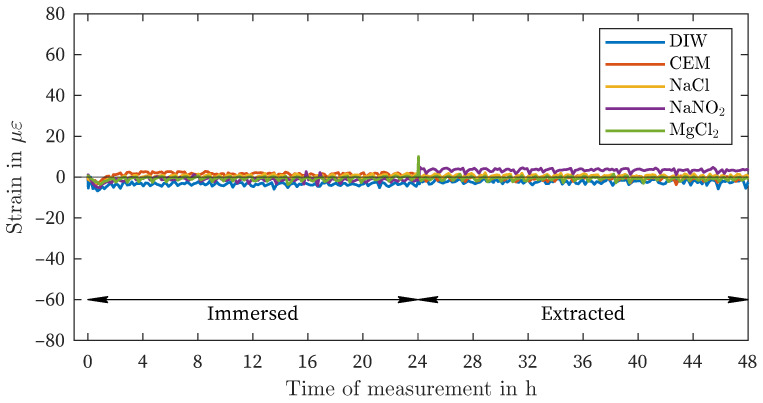
Fluid exposure, uncoated fiber.

**Figure 7 sensors-24-04659-f007:**
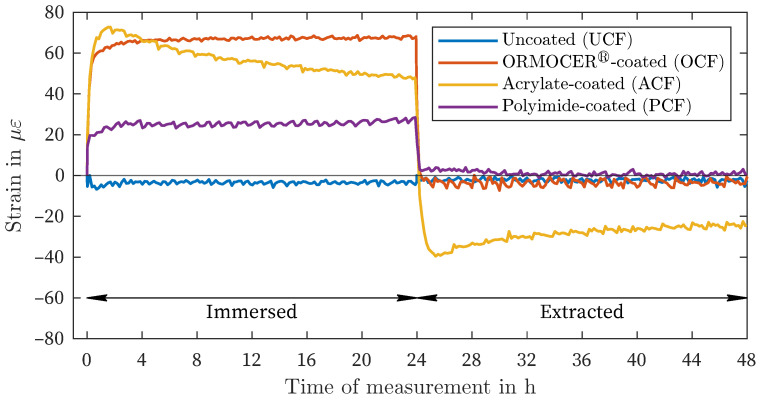
Fluid exposure, DIW.

**Figure 8 sensors-24-04659-f008:**
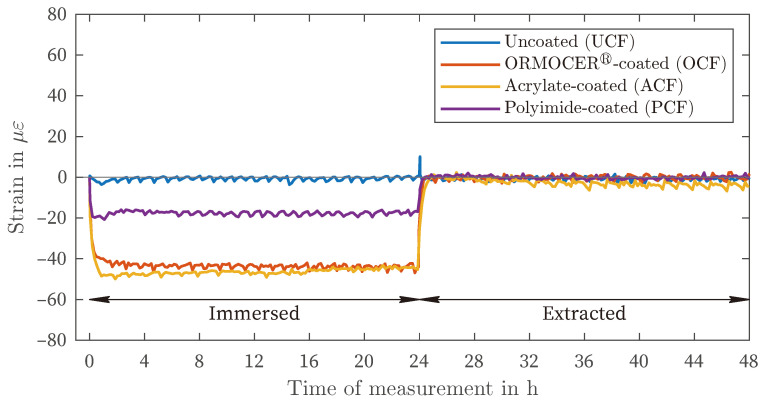
Fluid exposure, MgCl_2_ solution.

**Table 1 sensors-24-04659-t001:** Test liquids.

Liquid/Solution	Nomenclature	Concentration in g/L	Solubility ^1^ in g/L
Deionized water	DIW	-	-
Cement solution (CEM I 42.5 R)	CEM	-	-
Sodium cloride	NaCl	640	358
Sodium nitrite	NaNO_2_	1640	820
Magnesium chloride (MgCl_2_·6H_2_O)	MgCl_2_	3400	2350

^1^ at 20 °C.

**Table 2 sensors-24-04659-t002:** Fiber sensor characteristics.

Nomenclature	OCF	PCF	ACF	UCF
Coating	ORMOCER ^®^	Polyimid	Acrylate	Uncoated
Fibertype	LAL-1550-125	SM1550P	SMF-28e+	SMF-28e+
∅ Core in μm	9	9 (0.5)	8.2	8.2
∅ Cladding in μm	125 (1)	125 (1:3)	125 (0.7)	125 (0.7)
∅ Coating in μm	195	145 (5)	242 (5)	-
Attenuation in dB/km	<2.5	≤0.7	≤0.02	-
Strain coefficients in με/GHz	−6.67	−6.67	−6.67	−6.67

**Table 3 sensors-24-04659-t003:** Strain mean values, maximum amplitude and standard deviation of measurement.

Sensor	Value	Time	DIW	CEM	NaCL	NaNO_2_	MgCl_2_
		in hour	in με				
UCF	Mean	12–24	−4	1	1	−1	−1
		36–48	−2	−1	1	3	−1
	Std	12–24	0.7	0.7	0.8	1.0	0.9
		36–48	0.9	0.8	0.7	0.7	0.8
	Max	0–48	−3	5	10	−7	−4
OCF	Mean	12–24	67	64	18	2	−44
		36–48	−4	−2	−1	2	0
	Std	12–24	0.5	0.7	0.8	0.9	1.0
		36–48	1.4	1.3	1.4	1.1	1.1
	Max	0–48	69	66	19	4	−47
PCF	Mean	12–24	25	23	6	−1	−17
		36–48	1	0	−1	2	0
	Std	12–24	1.0	1.0	0.7	0.5	0.9
		36–48	0.8	0.6	0.8	0.7	0.7
	Max	0–48	28	27	8	4	−21
ACF	Mean	12–24	52	49	13	−1	−46
		36–48	−26	−27	−10	−2	−4
	Std	12–24	2.3	1.5	0.9	0.5	1.2
		36–48	1.3	1.3	1.0	1.2	1.3
	Max	0–48	73	73	19	−5	−50

## Data Availability

The datasets used and analyzed in the current study are available from the corresponding author on reasonable request.

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
