# Peer review of "The Impact of Liquids and Saturated Salt Solutions on Polymer-Coated Fiber Optic Sensors for Distributed Strain and Temperature Measurement"

_sensors, 2024, doi:10.3390/s24144659_

Round 1
Reviewer 1 Report
Comments and Suggestions for Authors
This article primarily investigates the application of fiber optic sensors in measuring concrete structures. Specifically, the researchers explored the impact of coating materials on strain and temperature measurements, with a focus on the expansion behavior of the coating materials in humid environments. The study's findings indicate that humidity can cause the coating materials to expand, potentially affecting the accuracy of strain and temperature measurements. Furthermore, the research revealed that factors such as temperature, pH , and salt content can also influence the expansion effects of the coating materials. The significance of this research lies in enhancing the reliability of fiber optic sensors for measuring concrete structures. For unprotected concrete infrastructure, exposure to humidity can distort strain and temperature measurements, leading to possible misinterpretations. By studying the expansion effects of coating materials, this research provides new guidelines for using fiber optic sensors in humid environments. Additionally, it offers potential for integrating fiber optic sensors for spatially resolved humidity detection. Overall, this study is crucial for improving the accuracy and reliability of concrete structure monitoring. However, there are areas that need improvement, with specific suggestions as follows:
1. The abstract mentions " significant discrepancies " and "negative deformation" but does not provide specific data or detailed descriptions of these results. Including quantitative data such as the range of deformation or specific negative deformation values would enhance the credibility and persuasiveness of the results.
2. The introduction references "hysteresis" but does not explain its meaning or specific application in this study. A brief explanation of this key term would help readers understand it more easily.
3. Mentioning some key experimental results or preliminary findings in the introduction could be useful for evaluating the study's initial conclusions and impact.
4. Although the experimental design mentions the effect of pre-treatment on coating expansion or contraction, it lacks detailed explanations. Adding detailed descriptions of the pre-treatment steps and their impacts would help readers understand the experimental results more comprehensively.
5. The experimental results could include a more in-depth explanation of certain phenomena. For instance, explaining why the acrylic coating fiber (ACF) exhibits different strain behaviors in certain solutions or why the strain change rate varies in different liquids would provide more detailed explanations or hypotheses.
6. The discussion mentions that the ACF's chemical hysteresis is not fully reversible within 24 or 48 hours. Has a longer observation period been considered, or are there hypotheses regarding long-term behavior?
7. In the discussion, detailed explanations of how Raoult's law and Donnan theory relate to polymer swelling and strain changes would be beneficial.
8. When discussing the uncertainty of the effects of high humidity or temperature pre-treatment, referencing results from similar studies would provide a deeper understanding.
Author Response
Thank you for your comments. I will respond to each of your comments below.
Comments 1: The abstract mentions " significant discrepancies " and "negative deformation" but does not provide specific data or detailed descriptions of these results. Including quantitative data such as the range of deformation or specific negative deformation values would enhance the credibility and persuasiveness of the results.
Response 1: Explanations have been added to the text.
Comments 2: The introduction references "hysteresis" but does not explain its meaning or specific application in this study. A brief explanation of this key term would help readers understand it more easily.
Response 2: An explanation has been added to the text.
Comments 3: Mentioning some key experimental results or preliminary findings in the introduction could be useful for evaluating the study's initial conclusions and impact.
Response 3: Explanations can be found in the Introduction.
Comments 4: Although the experimental design mentions the effect of pre-treatment on coating expansion or contraction, it lacks detailed explanations. Adding detailed descriptions of the pre-treatment steps and their impacts would help readers understand the experimental results more comprehensively.
Response 4: See comments 8, explanations are included in the text.
Comments 5: The experimental results could include a more in-depth explanation of certain phenomena. For instance, explaining why the acrylic coating fiber (ACF) exhibits different strain behaviors in certain solutions or why the strain change rate varies in different liquids would provide more detailed explanations or hypotheses.
Response 5: In our opinion, the results do not allow any conclusions to be drawn about the causes of the different deformation effects of the different coatings. The only conclusion that can be drawn from the results (comparison with UCF) is that the coating is the cause of the effect. Corresponding explanations are given in the text. Everything else, including the differences between the coatings, are hypotheses that are considered in the discussion. Again, there are explanations in the text.
Comments 6: The discussion mentions that the ACF's chemical hysteresis is not fully reversible within 24 or 48 hours. Has a longer observation period been considered, or are there hypotheses regarding long-term behavior?
Response 6: This aspect has already been taken into account in the discussion and conclusions and is recommended for further research activities.
Comments 7: In the discussion, detailed explanations of how Raoult's law and Donnan theory relate to polymer swelling and strain changes would be beneficial.
Response 7: We are of the opinion that detailed explanations of how Raoult's law and Donnan's theory describe the swelling effect are already available. References are also included.
Comments 8: When discussing the uncertainty of the effects of high humidity or temperature pre-treatment, referencing results from similar studies would provide a deeper understanding.
Response 8: The influence on fiber optic strain measurements described above has been discussed only to a limited extent in the literature, and relevant sources are listed in the bibliography. What has also not been considered is the influence of pre-treatment, which has already been described in the text. This is also a research result that we intend to investigate in a future study.
Reviewer 2 Report
Comments and Suggestions for Authors
This article conducts a series of experiments, and the relevant experimental results have certain reference significance for engineering practice. If appropriately revised, it is recommended to accept this article. However, the current version is given for rejection, and authors are encouraged to resubmit. Some problems or suggestions are as follow:
1. It seems that there is no graph showing the impact of humidity or pH value. If it exists, that would be great.
2. The article mentions the use of Rayleigh scattering - is it OFDR? Please provide a detailed explanation.
3. Can you provide the calibration relationship between the measured pH value and the actual pH value?
4. Please provide the measurement noise level for each graph to the best of your ability.
Author Response
Thank you for your comments. I will respond to each of your comments below.
Comments 1: It seems that there is no graph showing the impact of humidity or pH value. If it exists, that would be great.
Response 1: In the discussion and conclusion, this topic is mentioned as further research.
Comments 2: The article mentions the use of Rayleigh scattering - is it OFDR? Please provide a detailed explanation.
Response 2: Explanations and references have been added to the text.
Comments 3: Can you provide the calibration relationship between the measured pH value and the actual pH value?
Response 3: We measured the pH of the CEM solution before and after the measurement, for example to record any change caused by the CO2. In addition, the pH of the DIW is known (5.8). The change in CO2 in the air was also measured there. The basins are covered. Our pH meters are calibrated in different buffer solution (1,4,7,10,14).
Comments 4: Please provide the measurement noise level for each graph to the best of your ability.
Response 4: The noise of the available measurements is not useful. The data sheet of the measurement method with the corresponding characteristics is listed in the references. The variations shown in the diagrams are related to the temperature control of the climate device, as described in the text. In addition, changes due to hysteresis an climate control would be directly included in the calculation. The authors believe that Table 3 provides sufficient information to evaluate the measured values.
Reviewer 3 Report
Comments and Suggestions for Authors
This research article investigated the effects of moisture exposure, including different salt solutions and high pH, on DFOS using Rayleigh backscattering. Specifically, three fiber optic sensors with different coating materials and one uncoated fiber were exposed to five different solutions for 24 hours. The study showed significant differences in the deformation of the three coating types depending on the surrounding solution. The results of the study make it clear that such effects can impair measurement accuracy and lead to misinterpretations. At the same time, however, these effects could be used to carry out large-scale monitoring of chemical components using distributed fiber optic sensors.
The paper has potential, but still shows considerable potential for supplementation in order to be published in a Q1/Q2 journal. Specific references are given below.
- General note: The paper highlights the problems that can arise from DFOS measurements in different chemical environments. A proper solution to handle these effects is missing in the paper. This should be developed in order to increase the added value of the paper.
- The experimental design should be described more precisely. For example, in lin 88-89 it is not clear how the test stand and the measurement were realized. A real photo next to the principle sketch would be helpful.
- In line 119-120, the in-house calibration facility is used. It should be explained why this calibration is needed for the experiments and how this calibration works. Is there a standard calibration and does it differ from the in-house version?
- Line 123-124: The filtering process in the data analysis is not described, only mentioned very imprecisely. An explanation of what exactly was filtered out of the raw data signal and why filtering is necessary is required here.
- Line 177: What do the authors mean by chemical hysteresis? How can this be explained, especially if it only occurs with the ACF type? What is different from the other coating variants?
- Line 242-243: It should not only be stated that these results also affect the SHM, but also what should be done specifically to avoid these influences. It is assumed in this paper that the chemical influences are of a systematic nature. However, this assumption is made by testing one type of sensor at a time in one experiment. This hypothesis cannot be validated by this small number of experiments, especially if no repetition of the experiments was carried out to confirm the systematics behind these influences. There may also be a time variance to consider, which would make compensating for the effects in an SHM much more complicated. This is the main point of criticism of the study and should be made up for or emphasized more clearly.
- Chapter 4: How relevant is the influence of chemical parameters compared to temperature-induced or mechanically-induced strains? Is the effect in the same order of magnitude and if so, what must be done if all effects overlap. Then no differentiation is possible or damage detection in the SHM is no longer possible.
Comments on the Quality of English Language
Minor editing of English language required, small spelling mistakes, e.g., Table 2 "Fiber type", line 48 "an an".
Author Response
Thank you for your comments. I will respond to each of your comments below.
Comments 1: The paper highlights the problems that can arise from DFOS measurements in different chemical environments. A proper solution to handle these effects is missing in the paper. This should be developed in order to increase the added value of the paper.
Response 1: A proper solution has been added to the text.
Comments 2: The experimental design should be described more precisely. For example, in lin 88-89 it is not clear how the test stand and the measurement were realized. A real photo next to the principle sketch would be helpful.
Response 2: Corresponding explanations have been included in the text for better understanding. Since the fiber sensors are very difficult to see due to their small diameter, no photos of the experiments have been included. We think that Figure 1 already shows the experimental setup very well.
Comments 3: In line 119-120, the in-house calibration facility is used. It should be explained why this calibration is needed for the experiments and how this calibration works. Is there a standard calibration and does it differ from the in-house version?
Response 3: This is an unclear statement by the authors. The text has been changed accordingly.
Comments 4: Line 123-124: The filtering process in the data analysis is not described, only mentioned very imprecisely. An explanation of what exactly was filtered out of the raw data signal and why filtering is necessary is required here.
Response 4: This is a misunderstanding. The raw values were not changed in any way, they were just selected. The relevant part of the text has been changed.
Comments 5: Line 177: What do the authors mean by chemical hysteresis? How can this be explained, especially if it only occurs with the ACF type? What is different from the other coating variants?
Response 5: The text has been changed accordingly. The difference to the other coating variants is the polymer used. Explanations are included in the text.
Comments 6 Line 242-243: It should not only be stated that these results also affect the SHM, but also what should be done specifically to avoid these influences. It is assumed in this paper that the chemical influences are of a systematic nature. However, this assumption is made by testing one type of sensor at a time in one experiment. This hypothesis cannot be validated by this small number of experiments, especially if no repetition of the experiments was carried out to confirm the systematics behind these influences. There may also be a time variance to consider, which would make compensating for the effects in an SHM much more complicated. This is the main point of criticism of the study and should be made up for or emphasized more clearly.
Response 6: Further descriptions have been added to the text to make it clear that not just one sensor was tested in different liquids. Basically, the authors evaluated the strain values of 20 sensors. Including the preliminary tests not described in the text, many more sensors and liquids were tested. However, it is standard scientific practice to present only a subset of results from complete measurement campaigns. Suggestions for avoiding these influences within the SHM have been added to the text.
Comments 7: Chapter 4: How relevant is the influence of chemical parameters compared to temperature-induced or mechanically-induced strains? Is the effect in the same order of magnitude and if so, what must be done if all effects overlap. Then no differentiation is possible or damage detection in the SHM is no longer possible.
Response 7: The influence of the chemical parameters is relatively high compared to the temperature induced values. The temperature coefficient of 10µm/k was mentioned in the text. In comparison, the effect of temperature-induced OCF would be about 6-7 Kelvin. A suggestion for dealing with the findings regarding SHM has been added to the text.
Reviewer 4 Report
Comments and Suggestions for Authors
The authors demonstrate the impact of liquids and saturated salt solutions on the polymer-coated fiber-optic sensor. The paper has some interest for the field of high-sensitive fiber optic sensing field. There are some questions before its publishing in the high-standard Sensors journal.
1) In Fig. 1, how about the immersion length of fiber. In the experiment, the variables are the liquid and solution. However, the immersed fiber is in the bent state. How about this influence on the sensing test.
2) How about the response time when the fiber is immersed into the solution, or pulled out from the solution?
Author Response
Thank you for your comments. I will respond to each of your comments below.
Comments 1: In Fig. 1, how about the immersion length of fiber. In the experiment, the variables are the liquid and solution. However, the immersed fiber is in the bent state. How about this influence on the sensing test.
Response 1: The fiber sensor was immersed in the liquid over a length 10 cm. To prevent influences from edge effects, the evaluation area was limited to the middle 50 % of the immersed area. The bending radius is approximately 50 mm, which is well above the minimum bending radius for corresponding fiber sensors. A corresponding explanation was added to the manuscript.
Comments 2: How about the response time when the fiber is immersed into the solution, or pulled out from the solution?
Response 2: What is response time? The response of the coating to the liquids is included in the text and in the figure. If you mean response time from electrical engineering, I have added the measurement rate and sample rate to the text.
Reviewer 5 Report
Comments and Suggestions for Authors
This paper reports on the strain induced by different solutions, which is very important for real applications of distributed fiber sensing. It can be published once the following issues are addressed:
-
Line 48: The expression "an an other" needs to be corrected.
-
In Table 2: What is the meaning of the values in brackets?
-
It would be helpful to show the strain profile of the immersed fiber section and explain the start and end of the evaluation part. Is the evaluation length the same for all the fibers?
-
It is unclear whether the same fiber section was used repeatedly for different solutions or if new fiber sections were used for each solution.
-
Quasi-periodical strain variations can be observed in Figs. 3-7, even in the extracted phase. Could the author provide any explanation?
-
I am interested to know if there is any damage to the coating.
Author Response
Thank you for your comments. I will respond to each of your comments below.
Comments 1: Line 48: The expression "an an other" needs to be corrected.
Response 1: The text passage has been corrected accordingly.
Comments 2: In Table 2: What is the meaning of the values in brackets?
Response 2: The values in brackets in Table 2 are the uncertainties of the specified values.
Comments 3: It would be helpful to show the strain profile of the immersed fiber section and explain the start and end of the evaluation part. Is the evaluation length the same for all the fibers?
Response 3: All fiber sensor was immersed in the liquid over a length of 10 cm. To prevent influences from edge effects,
the evaluation length (same for all fibers) was limited to the middle 50 % of the immersed area. A corresponding explanation was added to the manuscript.
Comments 4: It is unclear whether the same fiber section was used repeatedly for different solutions or if new fiber sections were used for each solution.
Response 4:New fiber sensors were used for each test series in order to avoid influences from any hysteresis effects that might occur. A corresponding note was attached to the manuscript.
Comments 5: Quasi-periodical strain variations can be observed in Figs. 3-7, even in the extracted phase. Could the author provide any explanation?
Response 5: The quasi-periodic strain variations seen in Figures 3-7 are due to the climate control of the controlled aboratory as described in the manuscript (line 94 - 102. Care has been taken to minimize these effects, but they cannot be completely avoided. Since the measured values were not smoothed or otherwise processed, these strain variations from the climate control system are still visible.
Comments 6: I am interested to know if there is any damage to the coating.
Response 6: Scanning electron microscopy of the fiber sensors was not part of the study. However, such investigations are planned in follow-up studies.
Round 2
Reviewer 1 Report
Comments and Suggestions for Authors
The modification of the article meets my requirements and can be published.
Author Response
Comments: The modification of the article meets my requirements and can be published.
Response: Thank you for your comments and approval. Adjustments have been made regarding other review suggestions.
Reviewer 2 Report
Comments and Suggestions for Authors
Although the innovation of this article is not high enough, it does provide a referenceable experimental report for readers. If the authors had diligently addressed my questions, I would be inclined to recommend acceptance. However, it is regrettable that the authors did not respond to my inquiries with due seriousness. Furthermore, some of the responses themselves are quite questionable. Therefore, I can only suggest rejection.
In this article, the author states, “The slight fluctuations in strain are likely attributable to temperature and humidity control and within the range of measurement noise.” Moreover, the curve for NaNO2 in Figure 6 seems a bit strange. Thus, we inquired about the level of noise. The author responded, “The noise of the available measurements is not useful.” Certainly, noise is undesirable, but it must always be taken seriously. Even in exceptional cases where noise is not a concern, rigorous proof is necessary, not just a description of the experimental results. This is a very important and serious issue in the practical application of measurement fields.
Author Response
Comments:
Although the innovation of this article is not high enough, it does provide a referenceable experimental report for readers. If the authors had diligently addressed my questions, I would be inclined to recommend acceptance. However, it is regrettable that the authors did not respond to my inquiries with due seriousness. Furthermore, some of the responses themselves are quite questionable. Therefore, I can only suggest rejection.
In this article, the author states, “The slight fluctuations in strain are likely attributable to temperature and humidity control and within the range of measurement noise.” Moreover, the curve for NaNO2 in Figure 6 seems a bit strange. Thus, we inquired about the level of noise. The author responded, “The noise of the available measurements is not useful.” Certainly, noise is undesirable, but it must always be taken seriously. Even in exceptional cases where noise is not a concern, rigorous proof is necessary, not just a description of the experimental results. This is a very important and serious issue in the practical application of measurement fields.
Response:
Thank you for your feedback. The fact that we have some disagreement about the noise in our study does not mean that we do not take your comment seriously. We apologize for any misunderstanding. Appropriate changes have been made to the manuscript in response to your suggestions, and an explicit explanation is provided below.
The measurement noise in the present study is very complex. There are several reasons for this. In the case of an unloaded ideal fiber, i.e. the fiber is not subject to any environmental influences (temperature changes, humidity, vibrations, mechanical deformations), the measurement noise can originate from the following main sources: the measurement method itself (light source, detector, backscatter spectrum, spatial resolution, measurement rate), the splice quality (fiber connector, termination), the termination quality (bogging of the light signal), the fiber quality (variations in refractive index and attenuation due to the manufacturing process, Table 2) and the end face quality (fiber connector, bulkhead). If this fiber is stored in an environment under constant control (air conditioning), even with small variations in temperature and humidity (as shown in the results of this publication), it will also exhibit measurement noise and overlap with the above. In the third scenario, the fiber is immersed in the liquid, which may contain imperfections that cannot be excluded (saturation, temperature stability, evaporation). In addition, it is unclear whether the deformation effect caused by exposure to moisture can cause local differences in the coating and whether these are completely reversible after removal from the liquid.
Based on the above, the following explanation should be mentioned for the case in Fig. 6: It is possible that the taring of the fiber sensor before immersion took place exactly in the time range of a control process, so that a certain offset remains after removal. In addition, we cannot be certain that the coating was removed completely during the decoating process.
We hope to have addressed your comments and apologize again for the misunderstanding.
Reviewer 3 Report
Comments and Suggestions for Authors
Thank you for answering the questions. There is still one major point to be revised, which needs to be better incorporated into the manuscript.
Comments 6 Line 242-243: It should not only be stated that these
results also affect the SHM, but also what should be done
specifically to avoid these influences. It is assumed in this paper
that the chemical influences are of a systematic nature. However,
this assumption is made by testing one type of sensor at a time in
one experiment. This hypothesis cannot be validated by this small
number of experiments, especially if no repetition of the
experiments was carried out to confirm the systematics behind
these influences. There may also be a time variance to consider,
which would make compensating for the effects in an SHM much
more complicated. This is the main point of criticism of the study
and should be made up for or emphasized more clearly.
Response 6: Further descriptions have been added to the text to
make it clear that not just one sensor was tested in different liquids.
Basically, the authors evaluated the strain values of 20 sensors.
Including the preliminary tests not described in the text, many more
sensors and liquids were tested. However, it is standard scientific
practice to present only a subset of results from complete
measurement campaigns. Suggestions for avoiding these
influences within the SHM have been added to the text.
FollowUpComment 6: Thank you for this explanation. I cannot find any adjustment in the manuscript to the effect that more than one sensor of each sensor type was used in several solutions. My criticism is also not that 20 different sensor types (the authors have presented four sensor types here: OCF, PCF, ACF, UCF) should be presented. It is important that for each sensor type (OCF, PCF, ACF, UCF) several new sensors are tested in the respective solution in order to achieve statistical certainty about the statements made in the manuscript (reproducibility of the findings). If I have misunderstood this and the authors have carried out exactly these tests, then the authors should be able to give a mean value and a standard deviation for the strain curve in the respective solution in order to better support the statement of systematic deviations due to chemical reactions. This point must be incorporated into the manuscript so that the results appear valid.
Author Response
FollowUpComment 6: Thank you for this explanation. I cannot find any adjustment in the manuscript to the effect that more than one sensor of each sensor type was used in several solutions. My criticism is also not that 20 different sensor types (the authors have presented four sensor types here: OCF, PCF, ACF, UCF) should be presented. It is important that for each sensor type (OCF, PCF, ACF, UCF) several new sensors are tested in the respective solution in order to achieve statistical certainty about the statements made in the manuscript (reproducibility of the findings). If I have misunderstood this and the authors have carried out exactly these tests, then the authors should be able to give a mean value and a standard deviation for the strain curve in the respective solution in order to better support the statement of systematic deviations due to chemical reactions. This point must be incorporated into the manuscript so that the results appear valid.
Response:
Thank you for your feedback. This seems to be another misunderstanding. We used a new fiber for each solution. To that end, we have rewritten the manuscript to more clearly identify the test design. As the test design is extremely time and resource consuming, larger test campaigns are planned for further investigation. In addition, the results also show a certain pattern when comparing the respective coating materials.
We hope to have addressed your comments and apologize again for the misunderstanding.
Reviewer 5 Report
Comments and Suggestions for Authors
all the issues have been addressed
Author Response
Comments: all the issues have been addressed
Response: Thank you for your comments and approval. Adjustments have been made regarding other review suggestions.